# The rapid electrochemical activation of MoTe$_2$ for the hydrogen evolution reaction

Jessica C. McGlynn[1]*, Torben Dankwort[2], Lorenz Kienle[2], Nuno A.G. Bandeira [3,4], James P. Fraser [1], Emma K. Gibson [1], Irene Cascallana-Matías[1], Katalin Kamarás [5], Mark D. Symes [1], Haralampos N. Miras[1] & Alexey Y. Ganin [1]*

The electrochemical generation of hydrogen is a key enabling technology for the production of sustainable fuels. Transition metal chalcogenides show considerable promise as catalysts for this reaction, but to date there are very few reports of tellurides in this context, and none of these transition metal telluride catalysts are especially active. Here, we show that the catalytic performance of metallic 1T'-MoTe$_2$ is improved dramatically when the electrode is held at cathodic bias. As a result, the overpotential required to maintain a current density of 10 mA cm$^{-2}$ decreases from 320 mV to just 178 mV. We show that this rapid and reversible activation process has its origins in adsorption of H onto Te sites on the surface of 1T'-MoTe$_2$. This activation process highlights the importance of subtle changes in the electronic structure of an electrode material and how these can influence the subsequent electro-catalytic activity that is displayed.

---

[1] School of Chemistry, University of Glasgow, Glasgow G12 8QQ, UK. [2] Institute for Materials Science, University of Kiel, Kaiserstraße 2, 24143 Kiel, Germany. [3] BioISI–BioSystems and Integrative Sciences Institute and Centro de Química e Bioquímica, C8-Faculdade de Ciências da Universidade de Lisboa, Campo Grande, 1749-016 Lisboa, Portugal. [4] Centro de Química Estrutural, Instituto Superior Técnico, Universidade de Lisboa, Av. Rovisco Pais 1, 1049-001 Lisboa, Portugal. [5] Institute for Solid State Physics and Optics, Wigner Research Centre for Physics, Hungarian Academy of Sciences, P.O. Box 49Budapest 1525, Hungary. *email: jessicamcglynn112@gmail.com; alexey.ganin@glasgow.ac.uk

Transition metal dichalcogenides (TMDCs) show considerable promise as electrocatalysts for the hydrogen evolution reaction (HER)[1–4]. One of the most intriguing types of behaviour displayed by these materials is the propensity for certain TMDCs to undergo an *in operando* activation process during water electrolysis, leading to a gradual enhancement in electrocatalytic activity. $TaS_2$ and $NbS_2$ both exhibit such behaviour, which manifests as a gradual improvement in the essential metrics for the HER with sustained potential cycling (5000 cycles) to negative potentials[5]. However, so far, these are the only TMDC materials known to show such behaviour. The search for new TMDC electrocatalysts that also undergo *in operando* activation is therefore of fundamental interest and of critical importance for understanding the reaction mechanism of the HER and for the future development of novel electrocatalytic systems.

Recently, we[6] and others[7] reported that the monoclinic metallic phase of molybdenum telluride, $1T'$-$MoTe_2$ is a competent electrocatalyst for the hydrogen evolution reaction, albeit only capable of achieving a rather modest current density of $-10$ mA $cm^{-2}$ at overpotentials of around 360 mV in sulfuric acid[6,7]. This contrasts with the electrocatalytic activity displayed by the semiconducting $2H$-$MoTe_2$ polymorph, which was found to be largely catalytically inert. In our ongoing attempts to uncover the root cause of this difference in catalytic activity for the HER, we have now observed a remarkable and reversible *in operando* electrochemical activation of the nanocrystalline $1T'$-$MoTe_2$ material when it is held at cathodic potentials, which causes the overpotential required to sustain a current density of $-10$ mA $cm^{-2}$ to decrease from 320 to 178 mV. Intriguingly, if the reductive potential is removed, then the enhanced activity is lost and the overpotential required to sustain $j = -10$ mA $cm^{-2}$ reverts back to its original value, and the electrode must be re-activated by poising at a cathodic potential. The whole process occurs without any irreversible morphology or composition changes, suggesting that a change in electronic structure (as the HER progresses) is the underlying cause of the enhanced performance. In this paper, we firstly describe the nature of this electrochemical activation, before proposing a mechanism by which the enhanced catalytic effect is achieved.

## Results

### Electrochemical activation of nanocrystalline $1T'$-$MoTe_2$.
A nanocrystalline variant of $1T'$-$MoTe_2$ was synthesised via a solid-state route at a remarkably low temperature of 400 °C (experimental details are given in the Methods section and the Supporting Information). The nanocrystalline phase was fully characterised (PXRD, Raman spectroscopy, HRTEM/SAED, EXAFS, ICP-OES, XPS, Supplementary Figs 1–9 and Supplementary Discussion) and identified as phase-pure monoclinic $1T'$-$MoTe_2$, analogous to the previously reported crystalline material and nanoparticles[6,8]. The crystal structure of the monoclinic $1T'$-$MoTe_2$ is shown in Fig. 1a in comparison with the hexagonal $2H$-$MoTe_2$ phase (Fig. 1b).

The nanocrystalline material shows reasonable electrocatalytic behaviour when drop-cast onto a glassy carbon working electrode (Fig. 2a, red line) and thus, it tends to perform similar to current state-of-the-art catalysts (Supplementary Fig. 10 and Supplementary Table 1). However, continuous reductive potential cycling (over the range +0.2 V to −0.5 V vs. NHE at a scan rate of 100 mV/s) leads to an activation of the $1T'$-$MoTe_2$ phase as evidenced by a dramatic improvement in overpotential (blue line in Fig. 2a and Supplementary Fig. 11). At a current density of $-10$ mA $cm^{-2}$ (a benchmark proposed for the comparison of HER electrocatalysts[9–11]) the overpotential improves from $320 \pm 12$ mV to $178 \pm 8$ mV after only 100 cycles. After reaching this improved value of

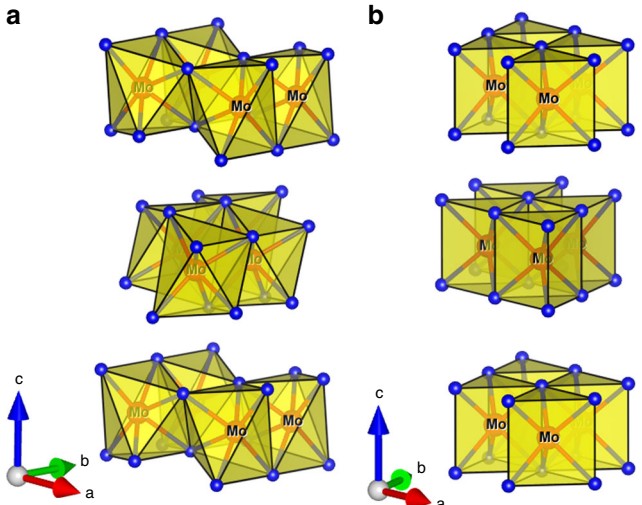

**Fig. 1** Crystal structure and $\{MoTe\}_6$ polyhedra showing the building blocks of each polymorph. **a** monoclinic $1T'$-$MoTe_2$ phase and **b** hexagonal $2H$-$MoTe_2$ phase

178 mV, the overpotential stabilises and no further changes are observed after 1000 cycles. A similar electrochemical enhancement can be achieved under a fixed potential bias: Fig. 2b shows that when a fixed overpotential of 320 mV is applied to the electrode, the current density increases rapidly over the following 90 min, eventually resulting in values in excess of $-35$ mA $cm^{-2}$. Fig. 2c and Supplementary Fig. 12 show that this enhanced current density corresponds to enhanced hydrogen production with increasing current density. Control experiments were also undertaken with a saturated $Hg/Hg_2SO_4$ reference electrode in order to rule out any silver leakage from the Ag/AgCl reference electrode as the source of the enhanced current densities obtained (Supplementary Fig. 13)[12].

The activation of nanocrystalline $1T'$-$MoTe_2$ is further evidenced by electrochemical impedance spectroscopy (EIS), which shows a reduction in charge transfer resistance ($R_{CT}$) by half after 100 cycles (Fig. 2d and Supplementary Fig. 14). Moreover, the reaction kinetics displayed a significant change upon potential cycling of nanocrystalline $1T'$-$MoTe_2$, with Tafel analysis after 100 cycles suggesting a possible change in the rate-determining step (Supplementary Fig. 15). TMDC electrocatalysts have been proposed to operate via the Volmer-Heyrovsky mechanism of hydrogen evolution, with the electrochemical desorption step (Heyrovsky) known to be slow, e.g., rate-limiting (Table 1):

The Tafel slope of nanocrystalline $1T'$-$MoTe_2$ undergoes an increase from $68 \pm 4$ mV decade$^{-1}$ when inactivated to $116 \pm 17$ mV decade$^{-1}$ after activation, thus indicating that the activation process results in the discharge (Volmer) step now being rate-limiting[13–15]. Upon cycling, there are also substantial changes in the exchange current density (from extrapolation of the Tafel slopes) and turnover frequency, with these parameters peaking at 0.503 mA $cm^{-2}$ and 0.12 s$^{-1}$ at 0 mV respectively after 100 cycles (Supplementary Table 2).

Interestingly, if the reductive potential is removed from an activated electrode (e.g., if the reductive potential cycling is ceased), even if only for a few seconds, then the overpotential for hydrogen evolution is found to revert back to essentially its original value (Supplementary Fig. 16). However, enhanced activity can be restored by simply re-applying a cathodic bias or by reductive potential cycling, both of which produce re-activation of the material on the same timescales as that seen with pristine material. This suggested to us that the cause of the

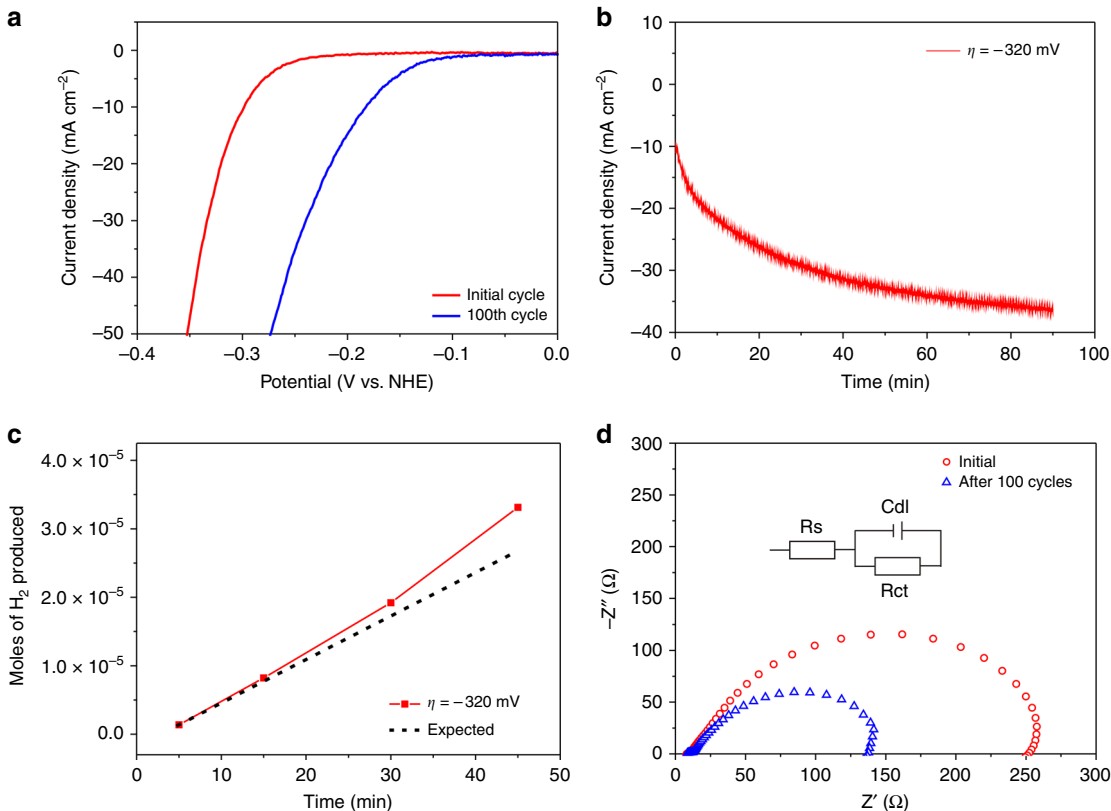

**Fig. 2** Electrochemical studies of nanocrystalline 1T'-MoTe$_2$. **a** Comparison of the current densities achieved by nanocrystalline 1T'-MoTe$_2$ before and after 100 cycles in 1 M H$_2$SO$_4$. Catalysts were prepared on a glassy carbon working electrode as described in the experimental section. Carbon felt and 3 M Ag/AgCl were used as the counter and reference electrodes, respectively. **b** Chronoamperometry profiles of the catalyst in 1 M H$_2$SO$_4$. The applied potential was chosen by initially performing LSV and choosing the potential corresponding to j = −10 mA cm$^{-2}$. Experiments were performed using a three electrode setup, with catalyst-deposited glassy carbon as the working electrode, 3 M Ag/AgCl as the reference and carbon felt as the counter electrode. **c** Representative trace of the number of moles of hydrogen produced with time as a constant potential is applied. The applied potential was chosen from LSV and corresponded to j = −10 mA cm$^{-2}$. Solid line indicates the experimentally determined value of hydrogen yield from gas chromatography for activated sample, while dashed line represents the theoretically calculated value without activation. **d** Nyquist plots showing electrochemical impedance spectroscopy on nanocrystalline 1T'-MoTe$_2$ before and after 100 cycles at −300 mV (vs. NHE). Uncompensated resistances were calculated as 8.1 and 8.3 Ω for disordered 1T'-MoTe$_2$ before and after 100 cycles, respectively. This corresponds well with the iR compensation function on the potentiostat which gave values of 9.8 and 10.1 Ω. The inset shows equivalent circuit model

| **Table 1 Underlying electrochemical processes occurring during the H evolution reaction** | | |
| --- | --- | --- |
| Volmer step | $H_3O^+ + e^- \rightarrow H_{ads} + H_2O$ | Discharge step |
| Heyrovsky step | $H_{ads} + H_3O^+ + e^- \rightarrow H_2 + H_2O$ | Electrochemical desorption step |
| Tafel step | $H_{ads} + H_{ads} \rightarrow H_2$ | Recombination step |

enhanced activity was likely to be electronic in nature, rather than related to irreversible (or slow) changes in the structure or morphology of the catalyst (see below for this analysis). In support of this hypothesis, comparison with nanocrystalline 2H-MoTe$_2$ shows no improvement in the overpotential after 100 reductive potential cycles (Supplementary Fig. 17), further suggesting that the origin of the activation process of 1T'-MoTe$_2$ is electronic in nature.

**Structural characterisation after electrochemical activation.** To test whether the enhanced catalytic activity could be caused by an irreversible crystal structure change, PXRD and Raman spectroscopy were performed immediately after reductive potential cycling. Both PXRD and Raman measurements were performed

directly on the surface of the glassy carbon electrode on which the catalyst had been deposited. This also allowed for confirmation that the preparation of the catalyst on the electrode had no effect on the structure of the materials. Fig. 3a and b show the PXRD patterns and Raman spectra of the nanocrystalline 1T'-MoTe$_2$ obtained before and after 1000 CV cycles. It is clear that the material remains unchanged; hence we can eliminate any structural changes as the cause of the enhanced catalytic activity.

Previous studies have suggested changes in morphology and/or composition as the origin of enhanced catalytic activity in TMDCs. For example, prior work on the metallic polymorph of TaS$_2$ attributed the improvement of the overpotential value after 5000 cyclic voltammetry cycles to substantial morphological changes after the cycling process[5]. To test whether any morphological changes were occurring in our case; we collected

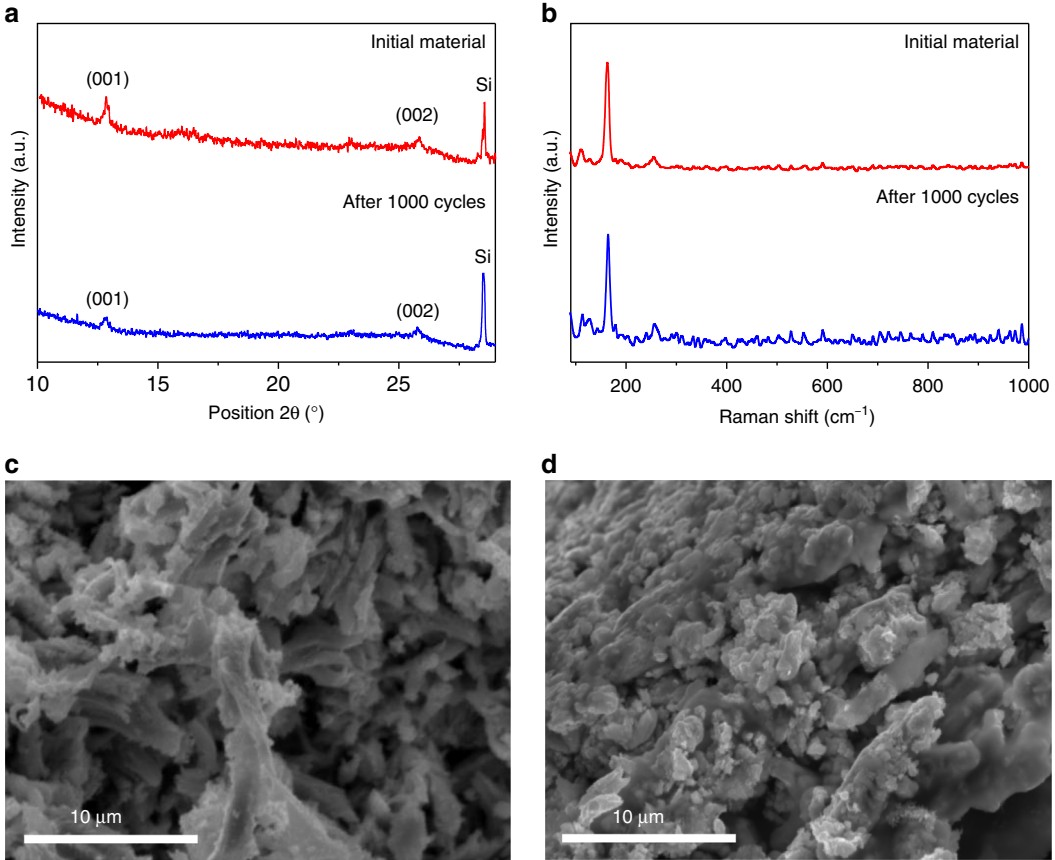

**Fig. 3** Stability studies of nanocrystalline 1T'-MoTe$_2$. **a** PXRD patterns of nanocrystalline 1T'-MoTe$_2$ before and after 1000 cycles. Patterns were measured directly on the surface of the glassy carbon working electrode on which the catalyst ink was deposited. A small amount of silicon powder was placed on the surface to act as an internal standard. PXRD patterns obtained after electrocatalytic measurements were performed directly on the electrode surface; therefore a layer of H$_2$SO$_4$ electrolyte is present and may result in weaker Bragg reflections. **b** Wide-range Raman spectra of nanocrystalline 1T'-MoTe$_2$ measured before (top) and after 1000 cycles (bottom). Raman spectra obtained after electrochemical measurements were performed directly on the electrode surface, therefore a layer of H$_2$SO$_4$ electrolyte is present thus resulting in an increased background after 1000 cycles. Particle morphologies of nanocrystalline 1T'-MoTe$_2$ before (**c**) and after cycling (**d**)

scanning electron microscopy (SEM) images of the material both before and after cycling. Fig. 3c shows that the microcrystalline morphology is preserved after cycling. Furthermore, EDX mapping after cycling shows Mo and Te to be well distributed throughout the material (Supplementary Fig. 18).

The electrochemically active surface area (ECSA) is also an important parameter to consider when evaluating the electrocatalytic activity of a material and can provide a very good indicator of substantial changes at the electrode interface[16,17]. The proportional relationship between the double layer capacitance ($C_{dl}$) and ECSA suggests that materials with the same values of $C_{dl}$ should have the same ECSA[18]. A higher ECSA after cycling could indicate that the enhancement of electrocatalytic activity was due to an increase in surface area upon cycling. However, the striking similarity of $C_{dl} = 3.17$ mF cm$^{-2}$ and 3.20 mF cm$^{-2}$ (for the material before and after cycling, respectively) rules out any significant improvement in catalytic performance due to a change in morphology or surface area (Supplementary Fig. 19).

An alternative explanation for the improved catalytic activity that needs to be excluded is the generation of new active sites (e.g., by generating tellurium vacancies through the leaching of cations/anions). In such a scenario, a change in catalyst composition would be expected. However, ICP-OES measurements show the stoichiometry of the nanocrystalline phase to be MoTe$_{1.97(5)}$ and MoTe$_{1.98(2)}$ before and after 100 cycles,

respectively (Supplementary Table 3). These results therefore prove that the stoichiometry remains constant with continuous cycling. The similarity in morphology before and after cycling thus rules out the effect of nanostructuring as the origin of enhanced catalytic activity (Fig. 3c).

The above results all suggest that an irreversible morphology and/or structural change in the catalyst is not the underlying cause of the enhanced catalytic performance displayed by 1T'-MoTe$_2$. Hence, to further investigate the underlying activation processes and to explore potential reaction pathways, we studied the electronic structure of these molecules and their intermediate adducts using density functional theoretical (DFT) calculations.

**Computational studies of potential reaction pathways**. Previous reports have shown that 1T'-MoTe$_2$ displays a high susceptibility towards electron doping on the plane surface by electrostatic gating, suggesting that it can accommodate excess electron density at the Fermi level[19–21]. It has been suggested that partially filled $d$-bands associated with the metallic character of 1T'-MoTe$_2$ allow for this additional electron density to be accepted during electrochemical reduction processes, giving rise to the HER activity of the metallic 1T'-MoTe$_2$ phase. Using this approach, the poor electrocatalytic activity displayed by the semiconducting 2H-MoTe$_2$ polymorph can be explained by its

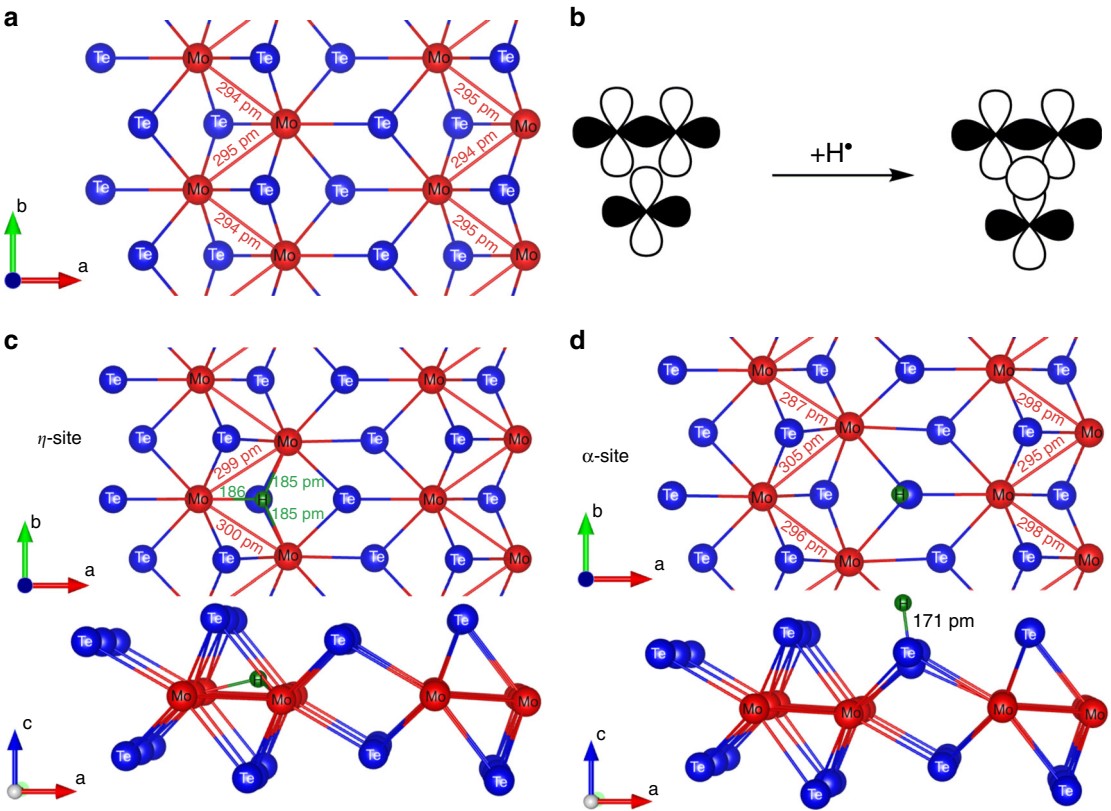

**Fig. 4** Computational studies of hydrogen adsorption on the surface of 1T′-MoTe$_2$. **a** The ab-plane projection of the optimised 2 × 2 unit cell of 1T′-MoTe$_2$ monolayer slab with selected calculated bond distances in pm (rev-PBE-D3/DZP + NO). **b** Schematic of frontier orbital symmetry with an H-adatom in the midpoint of the trinuclear bond axis. Optimised structures and bond distances (pm) of the regio-isomers of $^2_\infty[\text{MoTe}_2\text{H}_{0.125}]$ [(2 × 2 unit cell) with hydrogen adsorbed at (**c**) η-site and (**d**) α-site

inability to accept excess electron density[22]. Prior work by Seok et al., built on this framework to propose that the catalytic activity of 1T′-MoTe$_2$ arises from electron doping on the metallic surface, which in turn drives Peierls-type lattice distortion[7]. These authors suggested that it was the adsorption of hydrogen on Te active sites that was the underlying cause of this lattice distortion, resulting in a lower ΔG$_H$ and hence an active HER catalyst.

The production of hydrogen *in operando* and its likely highly reversible adsorption to the catalyst surface seems to correspond well with our experimental observations. Hence we chose hydrogen adsorption on Te as the starting point for our DFT analysis. To explain the nature of the adsorption process as hydrogen evolution progresses at the surface of 1T′-MoTe$_2$ and to probe deeper into the role of the hydrogen-ion adsorption on the electrochemical process, the 2 × 2 supercell of a monolayer slab was adapted as the working model of 1T′-MoTe$_2$ (Fig. 4a, Supplementary Table 4 and Supplementary Discussion for more details). The band structure and density of states reveal metallic behaviour with a Fermi level at −4.326 eV (ref. [18] and Supplementary Figs. 20–21). This behaviour is due to mild deviation of Mo atoms leading to shorter Mo–Mo bonds (295 pm) as the electronic structure analysis reveals. The recurring zig-zag arrangement of Mo atoms and the symmetry of the crystal frontier orbitals in the Fermi region (Supplementary Fig. 22) hint at the possibility of a μ$^3$-H bridging mode upon H atom insertion since there is sufficient nesting space for it (Fig. 4b). It is therefore worthwhile to explore the possibility of formation of a tri-haptic metal hydride which we called the η-site (Fig. 4c). We examined the viability and positions on the chemisorption energy scale for the three alternative regio-isomers (α, β, and ε) at a surface

coverage of 12.5% (in line with the coverage expected from the experimental values of the exchange current density). For the discussion of the regio-isomers of H-adsorption sites, we adopted the nomenclature proposed in a recent computational study on H-atom binding on 1T′-MoTe$_2$[7]. The main structural features of these isomers are compared with the proposed η-isomer as illustrated in Fig. 4d and Supplementary Fig. 23. The tri-haptic ligating mode shows a minimum on the potential energy surface attested by partial Hessian calculations yielding real values for the normal modes of the newly formed bonds. As predicted from qualitative arguments this new structure displays equal distances to all three metals (Supplementary Discussion). The energy of hydrogen uptake ΔE$_H$ = +0.58 eV is slightly lower (less endothermic) than on the α-site (Supplementary Table 5). The two remaining isomers (β and ε) display substantially higher ΔE$_H$, which is in line with the ligation energy trend reported before (Supplementary Fig. 24). Therefore, we restrict the following discussion to α-isomers and η-isomers only.

Since the η-isomer has a lower energy, one would expect an improvement in catalytic performance in a Te-deficient 1T′-MoTe$_2$ when additional edge sites and tellurium vacancies are created, similar to the mechanism proposed for the 1T-MoS$_2$ analogue[23]. To test this hypothesis, we synthesised and electro-chemically tested a nanocrystalline MoTe$_{1.8}$ compound. This MoTe$_{1.8}$ material does indeed demonstrate an *in operando* activation process (Supplementary Fig. 25), with the overpotential required to sustain a current density of −10 mA cm$^{-2}$ reducing from 320 to only 210 mV. The fact that this reduction in overpotential is not as marked as with the stoichiometric MoTe$_2$ provides useful information, as it suggests that the hydrogen

adsorption occurs preferentially on the tellurium sites, i.e., despite a higher $\Delta E_H$, the α-isomer is more favourable for hydrogen adsorption than the η-isomer. The preference for the Te-site indicates that it is the basal plane that is catalytically more active in 1T′-MoTe₂. In the case of the α-isomers, H adsorption leads to a change in Mo–Mo bond lengths and this may be the primary cause for the change in the experimental Tafel slopes on cycling (Supplementary Fig. 26). However, the change in bonding is minor and therefore, a rapid relaxation of the structure is expected in line with the observed experimental evidence, e.g., the overpotential reverts back to the original value when the potential bias is switched off. As suggested in the literature[7], the conventional volcano plot analysis[24] does not take into account potential secondary activation processes due to lattice distortion effects. Therefore, the relatively high value of $\Delta E_H$ for H adsorption on the α-site does not correspond to relatively high exchange current densities. Hence the computational studies suggest the distortion of the MoTe₂ structure as the primary cause of the activation process.

**Investigation into alternative routes of activation.** A plausible and alternative activation to the electron doping mechanism described above is the gradual removal of surface oxides which are present on the catalyst surface. It is possible that nanocrystalline 1T′-MoTe₂ is initially oxidized, and when subjected to cathodic bias the surface oxides are reduced and removed from the surface, thus revealing the true catalytic material. With this in mind, XPS studies were carried out under inert atmosphere in an attempt to observe any changes in oxidation state after cycling. Accordingly, Fig. 5a,b shows the high resolution 3d Mo and 3d Te

XPS spectra, respectively (Supplementary Figs. 27–28 for comparative spectra on samples exposed to ambient atmosphere). It is clear that there is only minor oxygen content after 100 CV cycles as evidenced by the broad shoulder which may indicate slight oxidation of the activated material. The peak shifts are within the resolution error of the instrument (0.1 eV) and are unlikely to be due to any substantial changes in oxidation state. Therefore, from this perspective, it is unlikely that the reduction of the oxidized surface could be the cause of activation (Supplementary Fig. 29).

In the same vein, one may argue that the proposed activation of the basal plane as presented in this work may also be valid for the edge sites. Therefore, the following discussion seeks to distinguish the role of each catalytic site. Since it has been well established that the edge sites are more easily oxidized than the basal plane[36], it is possible to differentiate between peaks which correspond to the oxidation of each site. Following a procedure by Bonde et al.[37], nanocrystalline 1T′-MoTe₂ was cycled at anodic potentials under a constant flow of nitrogen (Fig. 5c and Supplementary Fig. 30). Close examination of the anodic sweep reveals two distinct peaks corresponding to the oxidation of the edges (minor peak, maximum at +0.65 V (vs. NHE)) and the oxidation of the basal plane (major peak, maximum at +0.9 V (vs. NHE)). This correlates well with the literature work on MoS₂ by Bonde et al.[37]. In addition, a reduction wave (other than proton reduction) is observed at −0.24 V (vs. NHE). It should be noted that this reduction wave is always seen during the initial cycle of MoTe₂ and has been attributed to the reduction of surface oxides. As can be seen in Fig. 5c, by cycling nanocrystalline 1T′-MoTe₂ between −0.35 V and +1.05 V (vs. NHE), both edge sites and the basal plane are oxidized, resulting in a loss of catalytic activity and deactivation of the material.

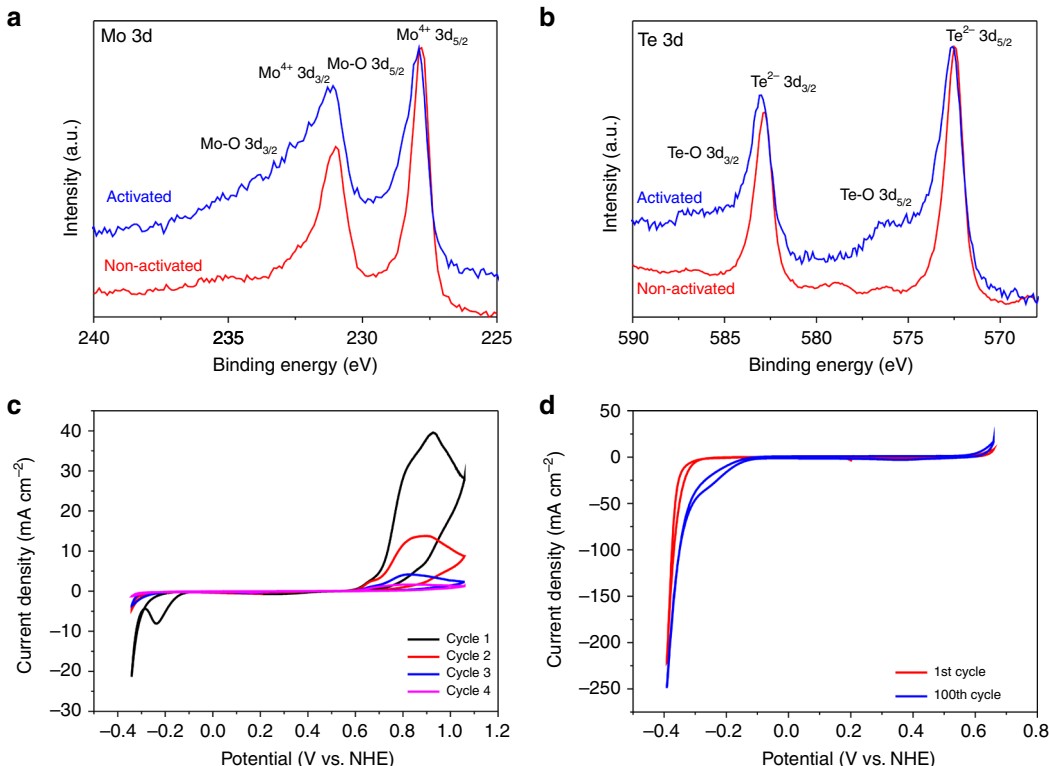

**Fig. 5** Investigation into the activation mechanism of nanocrystalline 1T′-MoTe₂. **a** 3d Mo XPS spectra of activated (after 100 cycles) and non-activated nanocrystalline 1T′-MoTe₂. **b** 3d Te XPS spectra of activated and non-activated nanocrystalline 1T′-MoTe₂. **c** Cyclic voltammograms showing the consequent deactivation of nanocrystalline 1T′-MoTe₂ in 1 M H₂SO₄ under a nitrogen atmosphere using a scan rate of 10 mV s⁻¹. During the first anodic sweep, two oxidations occur followed by a decrease in current density at cathodic potentials indicating catalyst deactivation. **d** Cyclic voltammogram of nanocrystalline 1T′-MoTe₂ before and after 100 cycles during which the edge sites are reversibly oxidized/reduced. Measurements were performed in 1 M H₂SO₄ under a nitrogen atmosphere with a scan rate of 100 mV s⁻¹

Subsequently, narrowing the potential range so that only the edges are oxidized, the effect of the intact basal plane can be analysed. Cycling nanocrystalline 1T′-MoTe$_2$ between $-0.4$ V and $+0.65$ V (vs. NHE) 100 times results in an identical overpotential shift from 320 mV to 175 mV (Fig. 5d). Thus, it is clear that oxidation of the edge sites has no impact on the overpotential improvement, and that the edges have no role in the activation mechanism. Further studies into the role of the edge sites are discussed in the Supplementary Discussion. On the other hand, when the potential range is extended to include oxidation of the basal plane, the catalyst becomes deactivated. This then indicates that the basal plane is responsible for the activation of nanocrystalline 1T′-MoTe$_2$.

## Discussion

In this work, we have developed a nanocrystalline variant of 1T′-MoTe$_2$ which maintains the metallic character associated with the 1T′-phase. By altering the morphology from crystalline to nanocrystalline, the material exhibits a dramatic enhancement in its electrocatalytic activity towards the hydrogen evolution reaction in acidic media. Using a range of analytical and characterisation techniques, we have shown that the origin of this enhanced electrochemical activity is not structural or morphological in origin. Instead, computational analysis suggests that electron doping under an applied reductive bias, and associated adsorption of hydrogen on certain Te sites in the material drives a distortion of the MoTe$_2$ structure which gives a more active catalyst as long as H remains adsorbed to the surface. This provides an explanation for the gradual improvement of catalytic activity over time (or with continued reductive cycling), as a gradual increase in the level of electron doping on the metallic surface leads to a gradual increase in hydrogen surface coverage until a limiting level of electron doping is reached at a given potential. It is significant that this enhancement effect occurs not at the relatively scarce edge-sites, but on the basal plane of the material. Hence, by activating the basal plane, the electrocatalytic performance of the bulk material can be vastly improved, providing an efficient route for optimising the HER activity of MoTe$_2$, and possibly other TMDCs as well.

## Methods

**Synthesis.** 1T′-MoTe$_2$ was prepared using a stoichiometric mixture of elemental molybdenum (Sigma Aldrich, 99.95%) and tellurium (Alfa Aesar, 99.999%). The powders were sealed under a vacuum pressure of $4.5 \times 10^{-2}$ mbar in a quartz ampoule which was carefully shaken in order to homogenize the mixture. The nanocrystalline phase was prepared by heating at 400 °C for 16 h before being allowed to cool to room temperature. The materials were reground and reannealed using the same heating protocols before being characterised. 2H-MoTe$_2$ was synthesised by reannealing 1T′-MoTe$_2$ at 700 °C (temperature ramp of 5 °C per min) for 24 h and cooling naturally to room temperature.

**Characterisation.** Powder X-Ray Diffraction (PXRD) was performed on a Panalytical Xpert-pro diffractometer with Cu K$_\alpha$ ($\lambda = 1.54178$ Å) operating in Bragg-Brentano geometry. For phase purity analysis, Raman spectroscopy was employed using a Horiba Jobin-Yvon LabRam Raman HR800 operating with a 532 nm laser. An aperture size of 100 μm and a 1% filter was used in order to prevent sample degradation. Scanning Electron Microscopy (SEM) images were obtained using a Phillips XL30 ESEM instrument coupled with an Oxford Instruments X-act spectrometer for EDX measurements. The EDX was calibrated using the INCA EDX software with Cu as the calibration standard. Transmission electron microscopy micrographs were obtained using a FEI Tecnai G2 F30 S-twin equipped with a 300 kV field emission gun. Diffraction pattern simulations were performed using JEMS[25]. ICP-OES was conducted using an Agilent 5100 calibrated to a range of known-concentration (ppm) solutions of Mo and Te. The BET surface area of 1T′-MoTe$_2$ was determined with a Quadrasorb Evo ODS-30. X-ray photoelectron analysis (XPS): 1T′-MoTe$_2$ samples were carefully packed and sent to the National EPSRC XPS Users' Service (NEXUS) at Newcastle University, UK and for measurements under inert atmosphere to the EPSRC National Facility for XPS ("Harwell XPS"). XPS spectra were acquired with a K-Alpha instrument (Thermo

Scientific, East Grinstead, UK), using a micro-focused monochromatic AlKα source (X-ray energy 1486.6 eV, spot size 400 × 800 microns). The emission angle was zero degrees and the pass energy was 200 eV for surveys and 40 eV for high resolution. Charge neutralization was enabled. The resulting spectra were referenced to the adventitious C 1s peak (285.0 eV) and were analysed using the free-to-download CasaXPS software package. Infrared (IR) measurements were performed using a Bruker 66v Fourier transform instrument with a deuterated tryglicine sulfate (DTGS) detector in the far-infrared region, and a mercury cadmium telluride (MCT) detector in the mid-infrared region, with 2 cm$^{-1}$ spectral resolution. Samples were ground and pressed into KBr pellets with 1 wt% concentration (~3 mg in 300 mg KBr) and measured in transmission mode. Time-of-flight powder neutron diffraction (PND) measurements were performed on the General Materials (GEM) diffractometer at the ISIS Facility of the Rutherford Appleton Laboratory, UK[26]. Samples were loaded in 0.7 mm diameter quartz capillaries and data were collected at room temperature. XAFS measurements were performed at the Mo K-edge on the B18 beamline at the Diamond Light Source, Didcot, UK. Data were collected in transmission mode (to $k_{max} = 18$) using a QEXAFS setup with a fast scanning Si(111) double crystal monochromator and ion chamber detectors, with a Mo foil placed between I$_t$ and I$_{ref}$. XAFS data processing was performed using IFEFFIT with the Horae package (Athena and Artemis)[27,28]. The amplitude reduction factor, S$_0^2$ was derived from EXAFS data analysis of the Mo foil.

**Electrochemical measurements.** All electrochemical measurements were performed using a Biologic SP-150 potentiostat with a single cell, three electrode setup in 1 M H$_2$SO$_4$ (unless otherwise stated). Catalyst powders were drop-cast onto a glassy carbon working electrode following the method of Gao et al.[29] This involved sonicating an ink consisting of 10 mg catalyst, 80 μL Nafion and 1 mL (3:1) water: ethanol for an hour. Thirty microlitre of the suspension was then drop cast on to the surface of the glassy carbon electrode (surface area 0.071 cm$^2$). Carbon felt and 3 M Ag/AgCl were used as the counter and reference electrodes, respectively. Identical experiments were repeated with a saturated Hg/Hg$_2$SO4 reference electrode in order to rule out any silver leakage as the source of improved performance. The electrode potentials were converted to the NHE scale using $E$(NHE) $= E$(Ag/AgCl) $+ 0.209$ V or $E$(NHE) $= E$(Hg/Hg$_2$SO$_4$) $+ 0.658$ V; and the ohmic resistances were compensated. Polarisation datasets were obtained using cyclic voltammetry at a scan rate of 100 mV s$^{-1}$. Tafel plots were obtained by linear sweep voltammetry with constant stirring at scan rates of 2 mV s$^{-1}$. Nyquist plots were obtained with frequencies ranging between 200 kHz and 1 mHz. The electrochemically-active surface area (ECSA) was obtained by sweeping the applied potential over a small potential range in the non-Faradaic region (in this case between 0.05 V and 0.25 V (vs. NHE)) at various scan rates (20, 40, 60, 80, 100, 120, 160, 200, 250, and 300 mV s$^{-1}$). Capacitive currents at 0.15 V (vs. NHE) were then plotted against scan rate, with the gradient being the double layer capacitance, C$_{dl}$. Due to their proportional relationship, ECSA can be estimated from the calculated C$_{dl}$ values[30]. Gas chromatography (GC) was performed using an Agilent GC 7890 A with a thermal conductivity detector. The GC system was calibrated using certified standards of hydrogen at various volume % in argon (CK Gas Product Limited (UK)) before use. For chronoamperometry, a potential of $-320$ mV (vs. NHE) was applied which corresponded to the overpotential required for $j = -10$ mA cm$^{-2}$. Faradaic efficiency measurements were obtained in a single airtight cell after being degassed under argon. A two electrode setup consisting of a platinum counter electrode and the catalyst-deposited glassy carbon working electrode was used for electrolysis. Galvanostatic electrolysis was then performed with a current density of 3.4 mA cm$^{-2}$ being applied. Twenty-five microlitre samples of the headspace were directly injected into the GC at various intervals. The Faradaic efficiency was then calculated as the ratio of expected H$_2$% in the headspace (as calculated from the charge passed) to the H$_2$% detected by GC measurements.

**Computational studies details.** The ADF-Band version 2017.113 software package was used for the in silico analysis of the catalyst hydrogenation process. The Perdew, Burke and Ernzerhof[31] generalised gradient approximate density functional as revised by Zhang and Yang[32] was employed throughout the calculations with the addition of Grimme's third generation dispersion correction (rev-PBE-D3). A combination of Herman-Skillman numerical atomic orbitals (NAOs) with triple zeta polarised (TZP) Slater type orbitals[33] were used for all the heavy atoms, whereas for hydrogen a double zeta polarised (DZP) augmentation was used. The geometry optimisations[34] of both coordinates and unit cell parameters were carried out in a 2 × 2 unit cell of 1T′-MoTe$_2$ single layer slab using a regular 1 × 3 k point grid. A partial Hessian calculation (involving the vibrational modes of the Mo–H bonds) was computed for $^2_\infty[\eta - \text{MoTe}_2\text{H}_{0.125}]$ yielding only real wavenumbers (with a lower bound 236 cm$^{-1}$ and upper bound 1291 cm$^{-1}$). The COSMO[35] dielectric continuum solvation scheme was utilised in all calculations with parameters for water. More accurate single point energy evaluations of the obtained stationary points were calculated using a denser 5 × 9 k point grid on the optimised unit cells.

The hydrogen atom addition was calculated with reference to the standard hydrogen electrode such that E(H$_2$) = E(H$^+$ + e$^-$); E symbolising electronic

energies, thus hydrogenation reaction energies were determined with the following formula $\Delta E_H = E(^2_\infty[MoTe_2H_{0.125}]) - E(^2_\infty[MoTe_2]) - 1/2E(H_2)$ where the last term is the electronic energy of hydrogen gas at the same level of theory.

## Data availability

The data sets generated during the current study are available from the corresponding authors upon reasonable request. A data set collection of electrochemical, PXRD, Raman raw data and optimised lattice coordinates from the DFT claculations are available at the University of Glasgow repository [https://doi.org/10.5525/gla.researchdata.892].

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

## Acknowledgements

We acknowledge the University of Glasgow, EPSRC (EP/P001653/1 and EP/R01308X/1), and the Carnegie Trust for a Research Incentive Grant (RIG007428) for supporting this work. M.D.S. thanks the Royal Society for a University Research Fellowship (UF150104). E.K.B. thanks the University of Glasgow for a Kelvin Smith Research Fellowship. N.A.G.B. acknowledges the Portuguese Foundation for Science and Technology for his fellowship (SFRH/BPD/110419/2015) and project grants (PTDC/QUI-QFI/29236/2017, PTDC/QUI-QFI/31896/2017, UID/MULTI/04046/2019, and UID/MULTI/00612/2019). The authors thank the Science and Technology Facilities Council for beam time allocation on GEM instrument at the ISIS Neutron Facility. X-ray photoelectron spectra were obtained at the National EPSRC XPS Users' Service (NEXUS) at Newcastle University, an EPSRC Mid-Range Facility and at the EPSRC National Facility for XPS ("Harwell XPS") operated by Cardiff University and University College London, under contract no. PR16195. We thank the Diamond Light Source for the award of beam time for the XAFS experiments as part of the Energy Materials Block Allocation Group SP14239. IR spectroscopy experiment was supported by the National Research, Development and Innovation Office (Hungary) grant No. FK 125063. T.D. and L.K. like to thank the federal state of Schleswig-Holstein, as well as the German research foundation, CRC 1261, project a6 for financial support.

## Author contributions

A.G. directed and coordinated the project. J.C.M. and A.G. designed the experiments, analysed, and interpreted the experimental data. J.C.M. and J.P.F. synthesised the samples with the help of H.N.M. J.C.M. performed electrochemical, S.E.M., X.R.D. and Raman measurements. T.D. an L.K. carried out and evaluated HRTEM experiments. N.A.G.B. performed computational studies. I.C.M. carried out XAFS experiments. E.K.G. interpreted the XAFS data. K.K. carried out IR spectroscopy studies. H.N.M. carried out ICP measurements. J.C.M., N.A.G.B., M.D.S. and A.G. wrote the paper. All authors commented on the paper.

## Competing interests

The authors declare no competing interests.
