## [Peer Review File · Nature Communications]

Reviewers' comments:

Reviewer #1 (Remarks to the Author):

This paper reports an interesting experimental observation that the catalytic performance of metallic 1T'-MoTe₂ is improved dramatically by electrochemical activation. Using a range of characterisation techniques, the authors have shown that the origin of this enhanced electrochemical activity is not structural or morphological in origin. Instead, computational analysis suggests that electron doping under an applied reductive bias, and associated adsorption of hydrogen on certain Te sites in the material drives a distortion of the MoTe₂ which gives a more active catalyst as long as H remains adsorbed to the surface.

Although the observation is interesting, I feel the proposed origin of the activation behavior is questionable and needs further studies. The authors claim it is because of the electron doping; however, the electron doping should be very fast, which contradicts the experimental fact that the activity increases gradually. I suggest the authors to perform more DFT calculations to study the kinetics of H adsorption etc and examine if they align with their proposed mechanism. Moreover, the paper would be easier to read if they did not break their figures into pieces.

Reviewer #2 (Remarks to the Author):

Report for "The rapid electrochemical activation of MoTe₂ for the hydrogen evolution reaction" by J. C. McGlynn et al.

The authors report that HER performances of nanocrystalline 1T'-MoTe₂ could be enhanced by reversible electrochemical activation, which is related to in-operando (volatile) hydrogen adsorption on the basal plane of nanocrystalline 1T'-MoTe₂. Many experimental evidences including XPS, EXAFS, XANES, TEM, SEM and XRD were used to support the authors' claims. However, this reviewer concerns 1) the actual performances of this sample compared with those of other state-of-the-art TMDC catalysts, 2) the novelty of the authors' interpretation considering former MoTe₂ works, and 3) the lack of experimental evidences for the improved HER operation on the basal planes. Details are described below. Therefore, this reviewer does not recommend this work for publication in Nature Communications.

1. The in-operando activation process of former TaS₂ and NbS₂ that the authors cited (ref. 5) attracted interests by their highly catalytic activity (e.g. overpotential of 50 mV for 10 mA/cm² with a loading density of 10 µg/cm²). Although the BET surface area in this work is similar to the value in ref. 5, this work reports 178 mV for 10 mA/cm², which is far below the previous report; before the activation, MoTe₂ shows better HER performances with TaS₂.

2. Then, the novelty or motivation of this work is not well supported for Nature Communications. According to the abstract and introduction, the motivation is few reports of tellurides (for HER) and poor catalytic performances of the tellurides. Considering the moderate HER performances described in '1', this reviewer is not convinced about the motivation why we need more studies with MoTe₂ about the activation using a similar mechanism explained in ref. 5. This work could be reported as another example of the electrochemical activation (described in ref. 5) in a more specialized journal.

3. How could the authors exclude the role of defects or edges for the improved HER? The low temperature (400 degree C) synthesis of MoTe₂ would make many unstable defects that can be modified during the HER process. And many edges are observed in TEM images (Fig. 3, S3). The authors should strengthen that those defects or edge geometries could not be the reason for the activation. Basal plane and DFT (Fig. 4) studies could be better conducted with single crystalline samples; the comparison experiment in Fig. S23 with single crystals does not show clear activation.

Moreover, the Hg/Hg₂SO₄ electrode control experiments shows little variance, and error range could be noted.

4. The synthesis part should be strengthened.

(1) The temperature (400 degree C) is quite low compared to former MoTe₂ synthesis studies. The melting temperature of Te is about 400 degree C, and many former studies mention that the 1T' is not stable. But in this work, the stability of the 1T'-MoTe₂ seems to be quite good. How can this stability be improved in this work?

(2) The EXAFS and XANES show a similar quality on the nanocrystalline and (typical) crystalline samples. But the XPS shows some oxidation. The apparently different observations could be explained (with the control experiments with the nanocrystalline and crystalline samples) because the surface area is the key for the HER.

Reviewer #3 (Remarks to the Author):

The manuscript by McGlynn addresses the hydrogen evolution reaction on MoTe₂. The authors contend that a key finding is that the overpotential of the material for HER can be lowered below 200 mV by the application of a cathodic bias. The authors do bring forward prior results for MoS₂ that show a similar effect, although the overpotential in the present case is more impressive. While I do not have any issues with the experiments and agree that the overpotential is very low, I do not entirely agree with the author's interpretation of the data.

1) The XPS provide in the supporting information suggest that there is significant oxygen on the catalyst surface, resulting in likely superficial Te oxidation. The effect of oxygen is complicated. Certainly a lot of prior studies have investigated the interaction of 1T MoS₂ with oxygen and found the introduction of defects (Nature Chemistry (2018), 10(12), 1246-125 and Materials Research Express (2017), 4(12), 125026/1-125026/9, to name a couple.) I would contend could affect HER. DFT calculations on the clean and pristine surface would potentially not capture all important aspects of the surface.

The authors need to address this. Also, perhaps a more likely scenario is that the application of the cathodic bias to activate the catalyst is reducing the concentration of Te-O species. Certainly the post-analysis XPS appears to show a decrease in the O-related features. Reduction of the catalytic surface might be supported by the data better than a lattice distortion (suggested by the DFT), since the former would likely show slower kinetics (Figure 2), consistent with the rather slow increase of HER activity as the cathodic bias is applied. The authors mention that "This provides an explanation for the gradual improvement of catalytic activity over time (or with continued reductive cycling), as a gradual increase in the level of electron doping on the metallic surface leads to a gradual increase in hydrogen surface coverage until a limiting level of electron doping is reached at a given potential." Is this just speculation with regard to the "gradual improvement." I do not see why this would be gradual. The authors should address these points.

2) Also the authors cycle their surfaces in some experiments to oxidizing potential that might put O on the surface. Do the authors use anoxic conditions for their experiments? If they have not, they might run such experiments.

3) The XRD after electrochemistry show weaker Bragg reflections. Is this do to a further disordering of the surface? There are also signal differences in the Raman data between initial and after spectra.

I believe that the authors need to rule out experimental issues that might influence their interpretation and modelling.

All of the changes we have made are outlined below (our responses are in Italic type). Where we have made changes to the manuscript or SI in response to the comments, we have highlighted these in yellow marker within the manuscript. We also clearly state whether anything is added to the manuscript in our response below. We believe that the changes made to the manuscript have improved it significantly, and we thank the referees for their help in this.

Reviewer1 (Remarks to the Author):

This paper reports an interesting experimental observation that the catalytic performance of metallic 1T'-MoTe₂ is improved dramatically by electrochemical activation. Using a range of characterisation techniques, the authors have shown that the origin of this enhanced electrochemical activity is not structural or morphological in origin. Instead, computational analysis suggests that electron doping under an applied reductive bias, and associated adsorption of hydrogen on certain Te sites in the material drives a distortion of the MoTe₂ which gives a more active catalyst as long as H remains adsorbed to the surface.

We thank the reviewer for insightful comments. As a minor comment we should point out that we did not discard the structural change and our computational data shows the minor change in Mo–Mo when tested with the 12.5 % coverage of the proposed a-site by adsorbed hydrogen.

Although the observation is interesting, I feel the proposed origin of the activation behaviour is questionable and needs further studies. The authors claim it is because of the electron doping; however, the electron doping should be very fast, which contradicts the experimental fact that the activity increases gradually. I suggest the authors to

perform more DFT calculations to study the kinetics of H adsorption etc and examine if they align with their proposed mechanism.

We agree with the referee that the activation process by the electron transfer would have been fast, however, only in the case of a molecular material. Due to the solid state nature of the MoTe₂ catalyst the initial adsorption of a H₃O⁺ to form adsorbed hydrogen required to drive structural change is limited by electrochemical processes at the surface such as diffusion and double layer formation. Hence, it is evident that a sufficient energy input is required to allow for H adsorption to occur. Therefore, we carried out additional DFT calculations to study how the amount of the adsorbed hydrogen on the α -isomer would change the energy of hydrogen adsorption for the adjacent α -site as the surface coverage increases. The graph below was added as Figure S27 to the SI (and reproduced for convenience below) shows the energy of H_{ads} required for adsorption on the first (12.5 % surface coverage), second (25 % surface coverage) and third (37.5 % surface coverage) α -site respectively. It is evident that contrary to expectation (as at higher coverage H–H becomes repulsive so the heat of adsorption would increase) in the case of MoTe₂ H_{ads} is decreasing until reaching the minimum at ca. 25 %, e.g. the situation when every other α -site is covered. We attribute it to minor structural change (as evident from the change in Mo–Mo distances) as the electron transfer is induced at high voltages.

Figure S27: The hydrogen adsorption energy on the α -site as a function of MoTe_2 surface coverage.

With regards to the gradual improvement of the activation process, this is unsurprising due to the sweeping nature of cyclic voltammetry used. If a sufficiently large potential was constantly applied, i.e. by chronoamperometry measurements, the activation would indeed be much quicker. However, constant application of large potentials results in the catalyst falling off of the glassy carbon substrate. Therefore, sweeping by cyclic voltammetry provides an alternative route to this activation which allows for the material to remain intact on the electrode surface. In this manner, choosing the potential range at which the catalyst is swept is key. Sweeping to sufficiently large potentials enables high current densities to be reached in a pulsing manner, and hence diffusion-limitation can be overcome all the while maintaining contact between the catalyst and substrate. Therefore, we found the optimum conditions for activation to be 100 cycles in the potential range of +0.2 V and -0.5 V (vs. NHE). Additionally, we found that narrowing the potential range (for example, sweeping between +0.2 V and -0.4 V (vs. NHE) rather than +0.2 V and -0.5 V (vs. NHE)) requires far more than 100 cycles in order for the improved overpotential of 178 mV to be reached. This indicates that the energy input / current density achieved controls the speed of activation, and explains why the observed improvement is gradual.

Reviewer 2 (Remarks to the Author):

1. The in-operando activation process of former TaS₂ and NbS₂ that the authors cited (ref. 5) attracted interests by their highly catalytic activity (e.g. overpotential of 50 mV for 10 mA/cm² with a loading density of 10 ug/cm²). Although the BET surface area in this work is similar to the value in ref. 5, this work reports 178 mV for 10 mA/cm², which is far below the previous report; before the activation, MoTe₂ shows better HER performances with TaS₂.

We thank the reviewer for the insightful comments. However, we must point out that, while nanocrystalline 1T'-MoTe₂ does not out-perform other state-of-the-art TMDC catalysts, it is the best performing transition metal telluride HER catalyst to date. In addition, state-of-the-art TMDC catalysts are most commonly sulfide based or variants of MoS₂, making a direct comparison between this work and ref. 5 unsuitable. The performance of nanocrystalline 1T'-MoTe₂ is, however, on par with these extensively studied TMDC catalysts. Hence, given the lack of literature on telluride based materials for the HER, we argue that nanocrystalline 1T'-MoTe₂ is an important addition to this library of TMDC electrocatalysts and warrants future study.

We have therefore added a table of current state-of-the-art HER catalysts for comparison in the supplementary information (Table S4), see also reproduced below for convenience.

Also, nanocrystalline 1T'-MoTe₂ is a bulk freestanding material, synthesised simply by stoichiometric reaction of the elements, without any additives or complex reaction schemes as required for current state-of-the-art catalysts. A catalyst of this calibre is therefore remarkable and we believe it to be of particular interest for publication in Nature Communications.

In addition, we would like to point out that the authors in Ref. 5 report the LSVs for H-TaS₂ and H-NbS₂. Both of these show the apparent activation of these materials to what would be extraordinarily low overpotentials (in the case of H-TaS₂, better than Pt). Pt has been shown

time and again to be almost the perfect HER electrocatalyst both experimentally and theoretically, so this result bears scrutiny, e.g. the assessment of the shape of the "optimised" catalysts, especially H-TaS₂ after 5000 scans. It is essentially a straight line, whereas all the other traces in the sequence are curves. This suggests over-compensation of the *iR* drop.

Alternative approach would be to investigate steady-state data in Ref. 5. Such data is provided in Figure S11A (Page 12 of the SI in the Ref. 5) for H-TaS₂. It shows the bulk electrolysis at the constant potential of 540 mV. The maximum current density reached after 24 h during the bulk electrolysis is 80 mA/cm². On the other hand, Figure S11B for LSV collected after 24 hrs suggests that this should require an overpotential of about 100 mV to achieve the current density of 80 mA/cm². However, 540 mV was applied in the bulk electrolysis. Therefore, the metrics reported in Ref 5 are vastly inflated and cannot be used as a reason to prevent publication of our paper as it stands.

Table S4: Comparison of HER activity in state-of-the-art catalysts

Catalyst	V (vs. NHE) at $j = 10 \text{ mA cm}^{-2}$	Tafel Slope (mV dec ⁻¹)	Reference
Double gyroid MoS ₂	0.28 V	50	a
MoS ₂ Nanoparticles	0.17 V	55-60	b
Exfoliated 1T-MoS ₂ Nanosheets	0.19 V	43	c
MoS ₂ /RGO	0.16 V	41	d
Amorphous MoS ₂	0.20 V	60	e
[Mo ₃ S ₁₃] ²⁻ nanoclusters on carbon supports	0.18 – 0.22 V	38-57	f
Cu ₇ S ₄ @MoS ₂ Hetero-nanoframes	0.13 V	48	g
Defect-rich MoS ₂ nanosheets	0.2 V	50	h
T-WS ₂ nanosheets	0.23	60	i
WS ₂ /RGO	0.26 V	58	j

References:

- a. Kibsgaard, J. *et al.* Engineering the surface structure of MoS₂ to preferentially expose active edge sites for electrocatalysis. *Nat Mater.* **11**, 963–969 (2012).

- b. Jaramillo, T. F. *et al.* Identification of Active Edge Sites for Electrochemical H₂ Evolution from MoS₂ Nanocatalysts. *Science* (80-.). **317**, 100–102 (2007).
- c. Lukowski, M. A. *et al.* Enhanced hydrogen evolution catalysis from chemically exfoliated metallic MoS₂ nanosheets. *J. Am. Chem. Soc.* **135**, 10274-10277 (2013).
- d. Li, Y. *et al.* MoS₂ nanoparticles grown on graphene: An advanced catalyst for the hydrogen evolution reaction. *J. Am. Chem. Soc.* **133**, 7296-7299 (2011).
- e. Benck, J. D. *et al.* Amorphous Molybdenum Sulfide Catalysts for Electrochemical Hydrogen Production: Insights into the Origin of their Catalytic Activity. *ACS Catal.* **2**, 1916-1923 (2012).
- f. Kibsgaard, J. *et al.* Building an appropriate active-site motif into a hydrogen-evolution catalyst with thiomolybdate [Mo₃S₁₃]²⁻ clusters. *Nat. Chem.* **6**, 248–253 (2014).
- g. Xu, J. *et al.* Ultrasmall Cu₇S₄@MoS₂ Hetero-Nanoframes with Abundant Active Edge Sites for Ultrahigh-Performance Hydrogen Evolution. *Angew. Chem. Int. Ed.* **55**, 6502-6505 (2016).
- h. Xie, J. *et al.* Defect-Rich MoS₂ Ultrathin Nanosheets with Additional Active Edge Sites for Enhanced Electrocatalytic Hydrogen Evolution. *Adv. Mater.* **25**, 5807-5813 (2013).
- i. Voiry, D. *et al.* Enhanced catalytic activity in strained chemically exfoliated WS₂ nanosheets for hydrogen evolution. *Nat. Mater.* **12**, 850-855 (2013).
- j. Yang, J. *et al.* Two-Dimensional Hybrid Nanosheets of Tungsten Disulfide and Reduced Graphene Oxide as Catalysts for Enhanced Hydrogen Evolution. *Angew. Chem. Int. Ed.* **52**, 13751-13754 (2013).

2. Then, the novelty or motivation of this work is not well supported for Nature Communications. According to the abstract and introduction, the motivation is few reports of tellurides (for HER) and poor catalytic performances of the tellurides. Considering the moderate HER performances described in ‘1’, this reviewer is not convinced about the motivation why we need more studies with MoTe₂ about the activation using a similar mechanism explained in ref. 5. This work could be reported

as another example of the electrochemical activation (described in ref. 5) in a more specialized journal.

With regards to the novelty of this work, we respectfully disagree with the reviewer's interpretation of the activation mechanism. In ref. 5, the authors report a substantial overpotential improvement generated by 'sample thinning', and prove that the morphology of their catalysts changes during cycling along with an increase in surface area. Hence, the origin of their 'activation' is clearly due to the substantial physical changes and damages to the NbS₂ and TaS₂ catalysts that occur during cycling. It is hard to argue in the current context how practical it is from the applications point of view. In contrast, we proved that in the case of 1T'-MoTe₂ the morphology remains intact and composition is unchanged (see Figures 3a-c in main text and Table S2 in supplementary information). Crucially, unlike the catalysts studied in ref. 5, we show that the original catalytic activity of 1T'-MoTe₂ is reversible, which further evidences that no physical changes / defect formation take place during activation (see Figure S11 in supplementary information). Therefore, this work describes an entirely different mechanism for the activation process from the work described in Ref. 5 and provides experimental evidences to back up the observation, the likes of which is entirely novel and relevant for Nature Communications.

3. How could the authors exclude the role of defects or edges for the improved HER?

The low temperature (400 degree C) synthesis of MoTe₂ would make many unstable defects that can be modified during the HER process. And many edges are observed in TEM images (Fig. 3, S3). The authors should strengthen that those defects or edge geometries could not be the reason for the activation.

The reviewer raises an interesting concern regarding the role of defects or edges as the reason for the enhanced activity. Concerning the generation of defects, we had performed ICP analysis on the materials before and after cycling, with both compositions being identical

within experimental error (see Table S2 in the supplementary information). Thus, the generation of defects with cycling can be ruled out as the source of increased activity. Likewise, due to the reversibility of activation, the generation of defects must be excluded since the original catalytic activity is returned after removal of potential bias, i.e. the generation of defects would result in a permanently improved overpotential. Since this is not the case, combined with ICP data, the role of defects can be excluded.

The reviewer also raises the question of the edge sites. Therefore, we further investigated the possibility of the edge sites being the cause of activation by following the procedure proposed in the classical paper: Bonde, J., Moses, P.G., Jaramillo, T.F., Norskov, J.K. and Chorkendorff, I. Hydrogen Evolution on Nano-Particulate Transition Metal Sulfides. *Faraday Discuss.* **140**, 219-231 (2008) – Added as ref. 37. This paper demonstrates that the edge sites are more easily oxidized than the basal plane (see additional reference: Kautek, W. and Gerischer, H. Anisotropic Photocorrosion of n-type MoS₂, MoSe₂ and WSe₂ Single Crystal Surfaces: The Role of Cleavage Steps, Line and Screw Dislocations. *Surf. Sci.* **119**, 46-60 (1982) – Added as ref. 36). Thus, it is possible to oxidize the edge sites and observe their effect (or lack of) on the catalytic activity. If the activity of nanocrystalline 1T'-MoTe₂ were to deplete, or remain stable after 100 cycles, this would confirm that the edges contribute to the activation. On the other hand, if the activation were to proceed as normal, we could confidently attribute the catalytic enhancement to be due solely to the basal plane. Accordingly, nanocrystalline 1T'-MoTe₂ was cycled between -0.35 V and +1.05 V (vs. NHE), see Figure 5c added to the manuscript also reproduced below for convenience. Close examination of the anodic sweep reveals two distinct oxidation peaks corresponding to oxidation of the edges (minor peak, maximum at +0.65 V (vs. NHE) and oxidation of the basal plane (major peak, maximum at +0.9 V (vs. NHE)). This correlates well with the literature work on MoS₂ by Bonde et al. Additionally, a reduction wave (other than proton reduction) is observed at -0.24 V (vs. NHE). It should be noted that this reduction wave is always seen during the initial cycle of MoTe₂ and has been attributed to the reduction of

surface oxides. As can be seen in newly added Figure 5C also reproduced below for convenience, by cycling nanocrystalline 1T'-MoTe₂ between -0.35 V and +1.05 V (vs. NHE), **both** edge sites and the basal plane are oxidized, resulting in a loss of catalytic activity and deactivation of the material.

Now, narrowing the potential range so that only the edges are oxidized, we see that the catalyst continues to be activated as normal, as evidenced by the increase in current density (new Figure S28 added and displayed for convenience below). Cycling nanocrystalline 1T'-MoTe₂ 100 times between -0.35 V and +0.65 V (vs. NHE) an overpotential shift from -320 mV to -175 mV at $j = 10 \text{ mA cm}^{-2}$ is observed (added Figure 5D and reproduced for convenience below), identical to the shift obtained with the edges remaining intact. Thus, since oxidation of the edges has no effect on the catalytic enhancement, it is unlikely that they are involved in the activation of nanocrystalline 1T'-MoTe₂. In addition to this, oxidizing the basal plane causes the catalyst to become deactivated. Hence, from this experimental evidence, it is clear that the activation is occurring on the basal plane sites.

Figure 5C: Cyclic voltammogram of the oxidation and consequent deactivation of nanocrystalline 1T'-MoTe₂ in 1 M H₂SO₄ under a nitrogen atmosphere with a scan rate of 10 mV s⁻¹. During the first anodic sweep, two oxidations occur followed by a decrease in current density at cathodic potentials indicating catalyst deactivation.

Figure S28: Cyclic voltammogram of the edge site oxidation of nanocrystalline 1T'-MoTe₂ and subsequent catalytic enhancement indicating the edges have no role in the activation process. Measurements were performed in 1 M H₂SO₄ under a nitrogen atmosphere with a scan rate of 10 mV s⁻¹.

Figure 5D: Cyclic voltammogram of nanocrystalline $1T'$ - MoTe_2 before and after 100 cycles whereby the edges sites are oxidized. Measurements were performed in 1 M H_2SO_4 under a nitrogen atmosphere with a scan rate of 100 mV s^{-1} .

Basal plane and DFT (Fig. 4) studies could be better conducted with single crystalline samples; the comparison experiment in Fig. S23 with single crystals does not show clear activation. Moreover, the $\text{Hg}/\text{Hg}_2\text{SO}_4$ electrode control experiments shows little variance, and error range could be noted.

We thank the reviewer for noticing the unclear activation presented here. Since we report the overpotentials achieved at $j = 10 \text{ mA cm}^{-2}$, we aim to show this improvement at low current densities. Therefore, the potential range could be narrowed to highlight the improvement in overpotential at low current densities. Hence we have amended both Figures S9 and S23 to highlight this activation, and therefore make our discussion clearer.

4. The synthesis part should be strengthened.

(1) The temperature (400 degree C) is quite low compared to former MoTe₂ synthesis studies. The melting temperature of Te is about 400 degree C, and many former studies mention that the 1T' is not stable. But in this work, the stability of the 1T'-MoTe₂ seems to be quite good. How can this stability be improved in this work?

The stability of 1T' over 2H at low temperature is not new and was demonstrated before by Schaak's group (Sun, Y. et al. Low-Temperature Solution Synthesis of Few-Layer 1T'-MoTe₂ Nanostructures Exhibiting Lattice Compression. Angew. Chem. Int. Ed. 55, 2830–2834 (2016). – Ref 4 in the SI) demonstrating preferential growth of 1T'-MoTe₂ over 2H-MoTe₂ phase at lower temperature. The underlying cause for structure stability proposed in Sun et al. paper is due to the very low energy difference required for transformation between two polymorphs. In addition, the authors speculated that nanostructuring stabilizes the structure due to “compressive lattice strain and polycrystalline nature”.

(2) The EXAFS and XANES show a similar quality on the nanocrystalline and (typical) crystalline samples. But the XPS shows some oxidation. The apparently different observations could be explained (with the control experiments with the nanocrystalline and crystalline samples) because the surface area is the key for the HER.

The reviewer is correct in their observation regarding the presence of oxygen in the XPS studies. The XPS of nanocrystalline 1T'-MoTe₂ was performed in ambient conditions; therefore the presence of oxygen is expected on the catalyst surface and is indeed seen in the XPS spectra before activation. Thus, from this perspective, it is possible that the activation may be due to the gradual removal of oxygen from the catalyst surface. We therefore carried out additional XPS studies under inert atmosphere to rule out the effect of oxygen at the EPSRC National Facility for XPS (the relevant acknowledgement was added

to the main text). Two nanocrystalline 1T'-MoTe₂ samples were measured: before activation and after 100 cycles. In both cases, the electrochemical cell, electrolyte and catalyst were degassed and maintained under a nitrogen atmosphere during the reaction. Before activation, the sample was immersed in 1 M H₂SO₄ and cycled 4 times, with the overpotential at $j = 10 \text{ mA cm}^{-2}$ reaching 320 mV. The second sample was cycled 100 times and the improved overpotential of ca. 180 mV at $j = 10 \text{ mA cm}^{-2}$ was obtained, analogous to experiments in ambient conditions. Both electrodes were immediately transferred under nitrogen to the glovebox, packed and sent for XPS studies, thus ensuring measurement in inert atmosphere. The high resolution 3d Mo and 3d Te XPS spectra are added to the main manuscript as Figures 5a and 5b and reproduced below for convenience. There is a very minor oxygen content present as evidenced by the broad shoulder in both unactivated and activated samples, but that this is essentially the same in both materials (or if anything the activated sample might be slightly more oxidised). The peak shifts are within the resolution error of the instrument (0.1 eV) and are unlikely to be due to any substantial changes in oxidation state. From this additional analysis, it is highly unlikely that the reduction of surface oxides is the cause of the activation. Additionally, we direct the reviewer to Table S2 in the supplementary information where ICP analysis rules out the generation of defects with cycling.

Figure 5: (a) 3d Mo XPS spectra of activated (after 100 cycles) and non-activated nanocrystalline 1T'-MoTe₂ under inert conditions.

Figure 5: (b) 3d Te XPS spectra of activated and non-activated nanocrystalline 1T'-MoTe₂ under inert conditions.

Reviewer 3 Remarks to the Author:

The manuscript by McGlynn addresses the hydrogen evolution reaction on MoTe₂. The authors contend that a key finding is that the overpotential of the material for HER can be lowered below 200 mV by the application of a cathodic bias. The authors do bring forward prior results for MoS₂ that show a similar effect, although the overpotential in the present case is more impressive. While I do not have any issues with the experiments and agree that the overpotential is very low, I do not entirely agree with the author's interpretation of the data.

1) The XPS provide in the supporting information suggest that there is significant oxygen on the catalyst surface, resulting in likely superficial Te oxidation. The effect of oxygen is complicated. Certainly a lot of prior studies have investigated the interaction of 1T MoS₂ with oxygen and found the introduction of defects (Nature Chemistry (2018), 10(12), 1246-125 and Materials Research Express (2017), 4(12), 125026/1-125026/9, to name a couple.) I would contend could affect HER. DFT calculations on the clean and pristine surface would potentially not capture all important aspects of the surface. The authors need to address this.

The reviewer is correct in their observation regarding the possible defects may be the possible origin for activation process. We considered this scenario and discussed it in the original version of the main manuscript (see the extract below). In fact, computational studies suggested that the presence of vacancies would be more favourable for HER. However, the experimental evidences suggest that the material with "synthetically" induced vacancies shows higher overpotential which led us to the conclusion that the activation does not happen due to formation of vacancies. To reemphasise this fact we slightly amended the statement in Page 8:

“Since the η -isomer has a lower energy, one would expect an improvement in catalytic performance in a Te-deficient 1T'-MoTe₂ when additional edge sites and tellurium vacancies are created, similar to the mechanism proposed for the 1T-MoS₂ analogue.²³ To test this hypothesis, we synthesised and electrochemically tested a nanocrystalline MoTe_{1.8} compound. This MoTe_{1.8} material does indeed demonstrate an in operando activation process (Fig. S22), however, with the overpotential required to sustain a current density of -10 mA cm^{-2} reducing from 320 to only 210 mV (instead of 178 mV in MoTe₂).”

Also, perhaps a more likely scenario is that the application of the cathodic bias to activate the catalyst is reducing the concentration of Te-O species. Certainly the post-analysis XPS appears to show a decrease in the O-related features. Reduction of the catalytic surface might be supported by the data better than a lattice distortion (suggested by the DFT), since the former would likely show slower kinetics (Figure 2), consistent with the rather slow increase of HER activity as the cathodic bias is applied.

The reviewer suggests an interesting theory. We have included and discussed a scenario that considers the reduction of the surface oxides as a possible mechanism (Figure 5A/ 5B and relevant discussion on page 9 of the main manuscript). Using the additional XPS studies conducted under inert atmosphere and on the product handled entirely under inert atmosphere (please see also the response to Reviewer 2 above for comprehensive discussion of the XPS studies) we demonstrated that only negligible surface oxidation exists before and after activation. There might be slightly higher surface oxidation in the activated material which strongly suggests that the activation due to the reduction of surface oxide is improbable.

In addition, one can argue that the XPS was not sensitive enough to detect the oxidation of edge sites and the activation could have happened due to the reduction of the oxide in the edge sites. However, it is possible to keep the edge sites oxidised anodically (see new Figures 5C/D and Figure S28 and discussion within the manuscript page 9-10) while keeping

the basal plane free from oxide but the activation still happens after 100 cycles. Therefore, since constant re-oxidation of the edge sites has no effect on the activation of 1T'-MoTe₂, in combination with the XPS analysis, we can confidently rule out the reduction of surface oxides (and edge oxides) as the cause of activation.

The authors mention that “This provides an explanation for the gradual improvement of catalytic activity over time (or with continued reductive cycling), as a gradual increase in the level of electron doping on the metallic surface leads to a gradual increase in hydrogen surface coverage until a limiting level of electron doping is reached at a given potential.” Is this just speculation with regard to the “gradual improvement.” I do not see why this would be gradual. The authors should address these points.

We thought carefully about this comment and as mentioned in response to Reviewer 1 we speculate that due to the solid state nature of the MoTe₂ catalyst the adsorption of a H₃O⁺ to form adsorbed hydrogen required to drive structural change is limited by electrochemical processes at the surface such as diffusion and double layer formation. Therefore, high potential (or current densities) is required to achieve activation. However, bulk electrolysis at high potential coincides with significant production of bubbles at the electrode. Therefore, we have chosen to carry out the activation by cyclic voltammetry. As we speculate that the reductive bias leads to electron doping we therefore re-phrase the relevant part of the main manuscript:

“Instead, computational analysis suggests that electron doping under an applied reductive bias, and associated adsorption of hydrogen on certain Te sites in the material drives a distortion of the MoTe₂ structure which gives a more active catalyst as long as H remains adsorbed to the surface. We speculate that the gradual improvement of catalytic activity over time (or with continued reductive cycling) leads

to a gradual increase in hydrogen surface coverage until a limiting level of electron doping is reached at a given potential.

2) Also the authors cycle their surfaces in some experiments to oxidizing potential that might put O on the surface. Do the authors use anoxic conditions for their experiments? If they have not, they might run such experiments.

We carried out the measurements under anoxic conditions: e.g., a constant flow of nitrogen and on the products handled under inert atmosphere, with the system being degassed prior to measurement. As can be seen from new Figure S29 added to SI (Reproduced for convenience below), a similar activation (within the error of measurements) occurs under anoxic conditions.

Figure S29: Comparison of the current densities achieved by nanocrystalline 1T'-MoTe₂ before and after 100 cycles in 1 M H₂SO₄ under ambient conditions (solid lines) and when handled under inert atmosphere (dashed lines). Catalysts were

prepared on a glassy carbon working electrode as described in the experimental section. Carbon felt and 3 M Ag/AgCl were used as the counter and reference electrodes, respectively.

3) The XRD after electrochemistry show weaker Bragg reflections. Is this do to a further disordering of the surface? There are also signal differences in the Raman data between initial and after spectra. I believe that the authors need to rule out experimental issues that might influence their interpretation and modelling.

Both PXRD and Raman spectroscopy are carried out directly on the electrode surface, therefore, after electrochemistry measurements there is a thin layer of 1 M H₂SO₄ present on the surface of the activated sample. For clarification, the captions of Figures 3a and 3b in the main text have been amended to include this oversight.

In order to carry out these measurements directly on the electrode surface, a bracket which specifically fits the glassy carbon working electrode was designed (Figure L). This involved careful mapping of the electrode surface area and alignment with the diffractometer. Hence, weaker Bragg reflections may be due to the layer of H₂SO₄ electrolyte on the surface and / or slight misalignment with respect of diffraction beam.

Figure A: *PXRD bracket designed to fit the catalyst-deposited glassy carbon working electrode.*

Reviewers' comments:

Reviewer #1 (Remarks to the Author):

The authors partially addressed my comments. However, I am still concerned about the origin of the active sites. It is a common practice to calculate the H adsorption free energy using computational H electrode model (doi: 10.1149/1.1856988), and use it to assess the catalytic activity (the closer to zero, the higher activity). Can the authors show a clear figure for it (like those in doi: 10.1149/1.1856988 and most other papers)? In fact, the Fig. S27 shows the H adsorption energy (not sure if it is calculated using computational H electrode mode or not) is high > 0.56 eV; for comparison, the Pt/MoS₂ edge has an $G(H^*)$ of < 0.1 eV. This casts doubts on the claimed active sites.

Reviewer #2 (Remarks to the Author):

Report for "The rapid electrochemical activation of MoTe₂ for the hydrogen evolution reaction" by J. C. McGlynn et al.

Overall, this reviewer's concerns are addressed well in the rebuttal letter: performance, novelty of the working mechanism, defects (edges), and synthesis.

Considering the novelty, the authors' approach and interpretation are (experimentally) related to 'ref. 5', and (theoretically) related to 'ref. 7'. Despite the partly overlapped stories, the reversible working (improvement of HER) mechanism has not been reported as the authors claimed. This reviewer feels that this work completes previous MoTe₂ HER studies (ref 7 in particular).

Therefore, this reviewer now recommends this work for publication in Nature Communications.

Reviewer #3 (Remarks to the Author):

I commend the authors in addressing the review in a serious way. I, however, still do not find the the rationale for the slow cathodic response convincing. The follow up experiments are useful, but I cannot still understand the slow cathodic response. Overall, however, the study is well done.

Response to Reviewers August 2019

All of the changes we have made are outlined below (our responses are in Italic type). Where we have made changes to the manuscript or SI in response to the comments, we have highlighted any new changes in yellow marker within the manuscript. We also clearly state whether anything is added to the manuscript in our response below. We thank the referees for their help in improving the manuscript.

Reviewer 1 (Remarks to the Author):

The authors partially addressed my comments. However, I am still concerned about the origin of the active sites. It is a common practice to calculate the H adsorption free energy using computational H electrode model (doi: 10.1149/1.1856988), and use it to assess the catalytic activity (the closer to zero, the higher activity). Can the authors show a clear figure for it (like those in doi: 10.1149/1.1856988 and most other papers)? In fact, the Fig. S27 shows the H adsorption energy (not sure if it is calculated using computational H electrode mode or not) is high > 0.56 eV; for comparison, the Pt/MoS₂ edge has an G(H*) of < 0.1 eV. This casts doubts on the claimed active sites.

We appreciate the comment since splitting the experimental details between SI and main manuscript may cause confusion. To afford clarification to the reader we have moved the section detailing the calculation of hydrogen adsorption from the supporting documents section to the main text. The changes are highlighted in yellow marker. It may be seen that the computational hydrogen electrode was adopted in the current calculations. Regarding the nature of the active site we apologise for the length of the discussion below and that it took us some time to respond to this query as we had to calculate the energy of adsorption on Pt and 1T'-MoS₂ as model systems to test the robustness of our calculations.

The active sites of 1T'-MoTe₂ have been established in the main manuscript by computing the adsorption energy of several sites. The energies of the sites are listed in Table S3 but we have now added a Figure S30 (similar to Figure 2 in 10.1149/1.1856988). These were also

compared with the energies in Pt and α -site of 1T'-MoS₂ that we calculated independently for proper comparison of the models. We also found that the values are in line with the literature. In the case of MoTe₂, the two lowest energy sites are α - and η - sites. Although theoretically the η -site, which involves the formation of a tri-haptic metal hydride, has a lower energy, experimental results suggest that the main active site is the α -site instead. This was determined experimentally by synthesising a tellurium deficient 1T'-MoTe_{1.8} material (Figure S22 and discussion in Page 8 of main manuscript). We also comprehensively investigated the possibility of the edge / defect sites as being the primary reason for the activity as discussed at length in the "Investigation of alternative routes for the activation process" section in page 9 of main manuscript. Thus, the α -site was deemed the main active site based on experimental evidences.

In this context, we agree with the referee that, conventionally, the closer the H adsorption free energy is to zero, the greater the activity of the catalyst. Therefore identifying the α -site as the most active site may appear as suboptimal in this case. However, this is in line with previous computational studies by Seok et al. (ref. 7 in main manuscript) in which the authors noticed very high positive energy of adsorption ΔE_H (eV) = 0.55 eV for α -site. At the same time, they observed very high exchange current densities which would place the catalytic activity of MoTe₂ out with the conventional Volcano plot according to Figure 1 in 10.1149/1.1856988. Their explanation was that Peierls -type lattice distortion leads to more favourable HER kinetics. We extended their model by proposing that the distortion can be further enhanced through the adsorption of H on α -site by reductive potential cycling. As presented in Figure S27 our calculations show that once H is adsorbed onto an α -site, the ΔE_H value of the neighbouring α -site becomes more favourable. We also observe a change in charge transfer resistance experimentally by electrochemical impedance spectroscopy. Figure 2d of the main manuscript shows R_{CT} to half in value upon activation of the material by reductive potential cycling, thus indicating that a more active material is obtained.

Again, we agree that ΔE_H (eV) of the activated material remains to be considerably large, i.e. >0.56 eV which can't explain the experimental current densities $\log(j_0) = -3.34$ A cm^{-2} calculated on geometrical area of the electrode of 0.071 cm^2 or $\log(j_0) = -4.80$ A cm^{-2} calculated based per BET area of 2 cm^2 (Page 14 in the SI for the calculations and Table S1 for the summary). However, we should point out that the exchange current densities are a result of extrapolation of the Tafel slope to 0 mV, thus the lower the Tafel slope, the lower the exchange current density. For example, platinum, has a Tafel slope of 30 mV dec^{-1} and the adsorption / desorption process follows the Volmer-Tafel mechanism in which the Volmer step is fast.

Volmer step	$\text{H}_3\text{O}^+ + \text{e}^- \rightarrow \text{H}_{\text{ads}} + \text{H}_2\text{O}$	Discharge step
Heyrovsky step	$\text{H}_{\text{ads}} + \text{H}_3\text{O}^+ + \text{e}^- \rightarrow \text{H}_2 + \text{H}_2\text{O}$	Electrochemical desorption step
Tafel step	$\text{H}_{\text{ads}} + \text{H}_{\text{ads}} \rightarrow \text{H}_2$	Recombination step

In other words the adsorption process (Volmer) on Pt is fast and it is reasonable to expect H^+ to “stick” to the surface which is reflected in negative values of ΔE_H .

Figure S30: Comparison of ΔE_H (eV) values at various H-bonding sites as listed in Table S3.

In the same vein, 1T-MoS₂ was reported to follow the Volmer-Heyrovsky mechanism (Tafel slope close to 40 mV dec⁻¹) in which again H⁺ adsorption step is fast and not rate limiting. One naturally should expect relatively low ΔE_H in this case in line with low experimentally observed exchange current densities (j_0). However, in the case of the activated 1T'-MoTe₂ we report a Tafel slope of 116 mV dec⁻¹. A Tafel slope of 120 mV dec⁻¹ is indicative of the Volmer mechanism in which hydrogen adsorption is being rate-limiting, hence in this manner the high ΔE_H value of the activated 1T'-MoTe₂ is unsurprising. Thus, the conventional volcano plot analysis, which is considered common practice, is limited due to sluggish H⁺ adsorption. We should also mention that non-activated 1T'-MoTe₂ with Tafel slope of 68 mV dec⁻¹ which probably corresponds to the Volmer-Heyrovsky mechanism (similar to 1T'-MoS₂) fits perfectly on the expected trend line (see the Table S5 and Figure below which represents graphical summary within the Table).

Table S5: Hydrogen adsorption energy relationship to the logarithm of the exchange current.

Surface	ΔE_H (eV)	$\log(i_0/\text{A}\cdot\text{cm}^{-2})$	Tafel slope (mV dec ⁻¹)	Ref.
$\alpha\text{-}\frac{\infty}{2}[\text{Mo}_8\text{Te}_{16}\text{H}]$	+0.67	-6.99	68	This work
$\alpha\text{-}\frac{\infty}{2}[\text{Mo}_8\text{S}_{16}\text{H}]$	-0.13	-4.90	48	23
$\frac{\infty}{2}[\text{Pt}_3\text{H}]$	-0.40	-3.34	30	12

Reviewer 3 (Remarks to the Author):

I commend the authors in addressing the review in a serious way. I, however, still do not find the rationale for the slow cathodic response convincing. The follow up experiments are useful, but I cannot still understand the slow cathodic response. Overall, however, the study is well done.

We thank the referee for the commendation, and we apologise for the lack of in-depth explanation regarding the gradual improvement of catalytic activity. To expand further, we have carried out additional measurements which explain our reasoning for the slow cathodic response (see below). Firstly, we agree that electron doping on the catalyst surface should be a fast process in the case of molecular materials; however, the MoTe₂ catalyst reported in this work is a bulk free-standing material, which raises the issue of limitation by electrochemical processes such as diffusion and double layer formation at the surface. These limiting electrochemical processes must be overcome in order to drive hydrogen adsorption; therefore a sufficient energy input must be supplied. Upon application of a sufficiently large potential, the catalyst will indeed be 'activated' quickly. For example, chronoamperometry measurements at a sufficiently large potential for a short time period would result in an activated material at a much faster rate. However, the large energy input required for this results in the vast production of hydrogen bubbles at the working electrode surface, which then causes the catalyst to lose contact with the glassy carbon substrate, and

hence the material falls off the electrode. Figure A / S31 (also reproduced below for simplicity) shows the resulting CV after one minute of chronoamperometry at a potential of -400 mV. After only one minute, the overpotential was found to improve to 220 mV at $j = 10$ mA cm⁻². The measurement, however, could not be continued for longer due to the increasingly large volume of hydrogen produced, which caused the measurement to cease.

Therefore, sweeping by cyclic voltammetry provides an alternative route to this activation, with the potential range (and number of cycles) being key parameters in achieving the improved overpotential. In this manner, a compromise is reached between the speed of activation and the energy input applied. Thus, due to the sweeping nature of cyclic voltammetry, the slow cathodic response is unsurprising. Sweeping to sufficiently large potentials enables high current densities to be reached in a pulsing manner, and hence diffusion-limitation can be overcome all the while maintaining contact between the catalyst and substrate. Therefore, we found the optimum conditions for activation to be 100 cycles in the potential range of $+0.2$ V and -0.5 V (vs. NHE). In support of this, the gradual improvement of catalytic activity can easily be seen with increasing cycle number (Figure B / S32). Further, narrowing the potential range at which the catalyst is swept provides additional evidence supporting the gradual improvement, as the material becomes only partially activated after 100 cycles, i.e. far more cycles are required when a lower energy input is applied (Figure C / S33). Therefore, we propose that the overpotential improvement is gradual since the speed of activation is controlled by the choice of experimental parameters, with potential range and number of cycles being the determining factors.

Figure A: Comparison of the current densities achieved by nanocrystalline 1T'-MoTe₂ before and after chronoamperometry measurements with an applied potential of -400 mV for one minute in 1 M H₂SO₄.

Figure B: Comparison of the current densities achieved by nanocrystalline $1T'$ - MoTe_2 after sweeping between the potential range of +0.2 V and -0.5 V (vs. NHE) for 25, 50 and 100 cycles in 1 M H_2SO_4 .

Figure C: Comparison of the current densities achieved by nanocrystalline $1T'$ - MoTe_2 after cycling the potential 100 times between +0.2 and -0.5 V (blue) and +0.2 and -0.4 V (wine) (vs. NHE) in 1 M H_2SO_4 .

REVIEWERS' COMMENTS:

Reviewer #1 (Remarks to the Author):

The authors have addressed my previous comments. A possible reason why the calculated formation energy of *H is apparently high is that the calculations are based on computational hydrogen electrode model, which assumes that the material is charge neutral. This is problematic because the material is charged during electrochemical reactions. A recent paper (DOI: 10.1021/jacs.8b03002) shows that the charge effect can greatly impact the calculated energetics for 2D materials. Although this method is computationally more expensive and it is not necessary to do for this paper, the authors are suggested to discuss the charge effect as a possible factor.

Reviewer #3 (Remarks to the Author):

The reviewers address the activation issue. It appears some experimental factors hampered the study. A rotating anode may of been advisable. Kinks in the polarization curves are likely due to bubble formation. In any case the authors offered their best experimental methods.

Reviewer 1 (Remarks to the Author):

The authors have addressed my previous comments. A possible reason why the calculated formation energy of *H is apparently high is that the calculations are based on computational hydrogen electrode model, which assumes that the material is charge neutral. This is problematic because the material is charged during electrochemical reactions. A recent paper (DOI: 10.1021/jacs.8b03002) shows that the charge effect can greatly impact the calculated energetics for 2D materials. Although this method is computationally more expensive and it is not necessary to do for this paper, the authors are suggested to discuss the charge effect as a possible factor.

*We thank the reviewer for their valuable critique regarding the state of the art methodology, and these new findings have now been inserted into the Computational Studies section of the Supplementary Information for the elucidation of the reader. Additionally, the suggested reference (Donghoon, K., Shi, J. & Liu, Y. Substantial Impact of Charge on Electrochemical Reactions of Two-Dimensional Materials. J. Am. Chem. Soc. **140**, 9127-9131 (2018)) has been included.*

Reviewer 3 (Remarks to the Author):

The reviewers address the activation issue. It appears some experimental factors hampered the study. A rotating anode may have been advisable. Kinks in the polarization curves are likely due to bubble formation. In any case the authors offered their best experimental methods.

The authors thank the reviewer for this suggestion. Indeed, the use of rotating disk electrodes would benefit our work greatly and we are aiming to purchase it for applications in our future studies.